# Investigating the mitochondrial genomic landscape of *Arabidopsis thaliana* by long-read sequencing

**Bansho Masutani**[1]*, **Shin-ichi Arimura**[2], **Shinichi Morishita**[1]

**1** Department of Computational Biology and Medical Sciences, Graduate School of Frontier Sciences, The University of Tokyo, Chiba, Japan, **2** Laboratory of Plant Molecular Genetics, Graduate School of Agricultural and Life Sciences, The University of Tokyo, Tokyo, Japan

* ban-m@g.ecc.u-tokyo.ac.jp

**Data Availability Statement:** All the source code, documentation, the results on the real-datasets, scripts to generate synthetic datasets, and validation pipelines are available at https://github. com/ban-m/reconstruct_mito_genome. The whole genome sequencing data are available at Sequence

## Abstract

Plant mitochondrial genomes have distinctive features compared to those of animals; namely, they are large and divergent, with sizes ranging from hundreds of thousands of to a few million bases. Recombination among repetitive regions is thought to produce similar structures that differ slightly, known as "multipartite structures," which contribute to different phenotypes. Although many reference plant mitochondrial genomes represent almost all the genes in mitochondria, the full spectrum of their structures remains largely unknown. The emergence of long-read sequencing technology is expected to yield this landscape; however, many studies aimed to assemble only one representative circular genome, because properly understanding multipartite structures using existing assemblers is not feasible. To elucidate multipartite structures, we leveraged the information in existing reference genomes and classified long reads according to their corresponding structures. We developed a method that exploits two classic algorithms, partial order alignment (POA) and the hidden Markov model (HMM) to construct a sensitive read classifier. This method enables us to represent a set of reads as a POA graph and analyze it using the HMM. We can then calculate the likelihood of a read occurring in a given cluster, resulting in an iterative clustering algorithm. For synthetic data, our proposed method reliably detected one variation site out of 9,000-bp synthetic long reads with a 15% sequencing-error rate and produced accurate clustering. It was also capable of clustering long reads from six very similar sequences containing only slight differences. For real data, we assembled putative multipartite structures of mitochondrial genomes of *Arabidopsis thaliana* from nine accessions sequenced using PacBio Sequel. The results indicated that there are recurrent and strain-specific structures in *A. thaliana* mitochondrial genomes.

## Author summary

Plant mitochondria have genes with important functions. For example, some mitochondrial genomes contain a gene responsible for cytoplasmic male sterility, a phenotype that is unable to create mature pollen. However, despite their small sizes, plant mitochondrial

Read Archive (accession numbers ERR3415817-ERR3415831), DNA Data Bank of Japan(accession number: DRA010390), and PacBio's official repository (URL is https://downloads.pacbcloud.com/public/SequelData/ArabidopsisDemoData/).

**Funding:** This work was supported by JSPS KAKENHI under Grant Number 16H06279 (to S.M. and S.A), and AMED (Japan Agency for Medical Research and Developmentunder) Grant Number 20gm1110007h0003 (to S.M.). The funders had no role in study design, data collection and analysis, decision to publish, or preparation of the manuscript.

**Competing interests:** The authors have declared that no competing interests exist.

genomes can be difficult to assemble even if we use state-of-the-art long-read sequencers. The main obstacle is their high structural diversity and low sequence diversity, which hamper traditional methods to assemble plant mitochondrial genomes. Here, we introduce a new method for grouping long-reads to individual structures. For this purpose, we explored two traditional models in sequence analysis; hidden Markov model and partial order alignment, which enable us to detect a single base variation among several thousand bases and output accurate clusters while managing with observation errors associated with long-read sequencing. Applying this method to nine PacBio Sequel read datasets from *Arabidopsis thaliana*, we uncovered putative but unknown structures of plant mitochondrial genomes, suggesting that strain-specific structures are present in mitochondrial genomes, and that linear DNA fragments appear repeatedly in several strains.

## Introduction

Since Lynn Margulis confirmed with substantial evidence that mitochondria originated from external bacteria [1], genomes of mitochondria, which are called mitogenomes, have been extensively investigated.

We now know that the mitogenomes of plants differ from those of animals. For example, plant mitogenomes comprise hundreds of thousands to millions of nucleotides [2]. Additionally, mitochondria have a gene that is responsible for cytoplasmic male sterility, a phenotype in which the individual cannot produce mature pollen [3]. Because of this unique effect on fertility, plant mitogenomes should be studied independently.

The first complete plant mitogenome came from *Arabidopsis thaliana*, using traditional Sanger sequencing [4]. Since the emergence of Illumina sequencing technology, the mitogenomes of many species (e.g., rice) have been assembled as circular contigs [5]. Long-read sequencers released by PacBio or Oxford Nanopore Technology have accelerated this process [6–15]. Consequently, many studies have determined the nucleotide sequences of plant mitogenomes, compared the resulting assemblies, and proposed evolutionary scenarios for current patterns [11], [16].

Nonetheless, there is room for improvement in terms of obtaining a more in-depth understanding of mitogenome assemblies. One area that requires further clarification is the phenomenon of "alternative reality" [13]–i.e., plant mitogenomes are not static but change dynamically, and nucleotide sequences can vary due to recombination among repetitive sequences [2, 17–19].

This recombination presumably combines DNA fragments together, leading to the creation of many configurations (multipartite structures) in plant cells. Although this phenomenon was observed in the study that first published a plant mitogenome assembly [4], plant mitogenomes are usually represented as "master circles"–i.e., single circular lists of nucleotides. Assays to reveal these multipartite structures based on polymerase chain reaction (PCR) are also not very useful, as artifacts of PCR-mediated recombination may be created [20].

Importantly, Kozik *et al.* recently presented pioneering work that accounted for this insufficient representation [13]. They first constructed an initial assembly, then "tiled" long reads along the contigs to find the building blocks of the multipartite structures. Although this approach recovered the putative multipartite conformations, plant mitogenomes may be further elucidated by assembling each structure separately like a pan-genome approach.

To this end, it is necessary to group reads into individual structures by using variations among structures. An algorithm for solving this problem based on the Minimum Error

Correction Problem has been reported [21]. This algorithm runs in $O(s2^k)$ time, where $s$ and $k$ are the number of variant sites and the maximum coverage of variant sites, respectively. However, this algorithm is not practical because its runtime has an exponential relationship with maximum coverage ($k$), which is typically more than 600-fold in mitogenomes, and because it does not consider structural variations among multipartite structures. Tools for clustering long reads have been published [22], [23]; however, these programs were generally designed to group cDNA or RNA reads into alternative splicing variants rather than to cluster multipartite structures. Therefore, another highly efficient method is needed.

In this study, we aimed to resolve issues related to the high error rate ($\approx 14\%$) associated with long reads and the low divergence among mitogenomes, to group reads into individual structures.

## Materials and methods

It may be useful to recall *K*-means clustering. In this method, we iteratively compute the centers of clusters and assign each point to one of the nearest centers, as in Algorithm 1:

**Algorithm 1** *K*-means clustering

```
Input: A set of real numbers V = {x₁, ···, x_N} and the number of
clusters K
Output: Assignment of cluster ID w_n ∈ {1, ···, K} to ∀x_n ∈ V
 1: Assign each point to a cluster randomly.
 2: repeat
 3:    C_k ← {x_n | w_n = k} for each k          ▷Points in the k-th cluster.
 4:    M_k ← ∑_{x∈C_i} x/|C_i| for each k        ▷Compute centroid M_k of C_k
 5:    w_n ← arg max_k − ‖M_k − x_n‖² for each n  ▷Assign each point x_n to
   the nearest centroid.
 6: until Convergence
 7: Return w₁, ···, w_N
```

To use this algorithm to group long reads into corresponding clusters, we need to define a model to represent a cluster and determine the similarity measure between a read and a cluster to update the assignment. In the case of *K*-means clustering, we represent a cluster by its center, and the similarity measure is the negative Euclidean norm. In the following section, we first describe the model and then move on to the similarity measure.

### POA for representing multipartite structures

As noted in the Introduction, plant mitochondrial assemblies are currently characterized as circular contigs known as "master circles," which represent the major configuration of the mitogenomes. Thus, we can align reads to the reference while allowing a certain amount of mismatches, deletions, insertions, and structural variations.

From these alignments, we can build a model by integrating reads into the aligned locations in the reference master circle. One approach is to regard the master circle as an initial graph and employ POA [24] to merge reads one by one, resulting in a single POA graph.

However, as shown in the following section, it takes a long time–$O(R^2 N)$ time–to construct a POA graph from $N$ reads if these reads are sufficiently similar, where $R$ is the length of the longest read. Thus, we want the graph to be as small as possible for computational efficiency.

To this end, we adopted a similar approach as in Racon [25]. Specifically, we split the reference into sections of $LU$ bp (for example, 2K bp) length. We carry out clustering procedure on each of them and merge consecutive sections to obtain final clustering. To this end, for each section, we collected all the read sub-sequences aligned to it and split these sequence into smaller chunks with $U$-bp (for example, 100 bp) length. Then, these chains of aligned segments

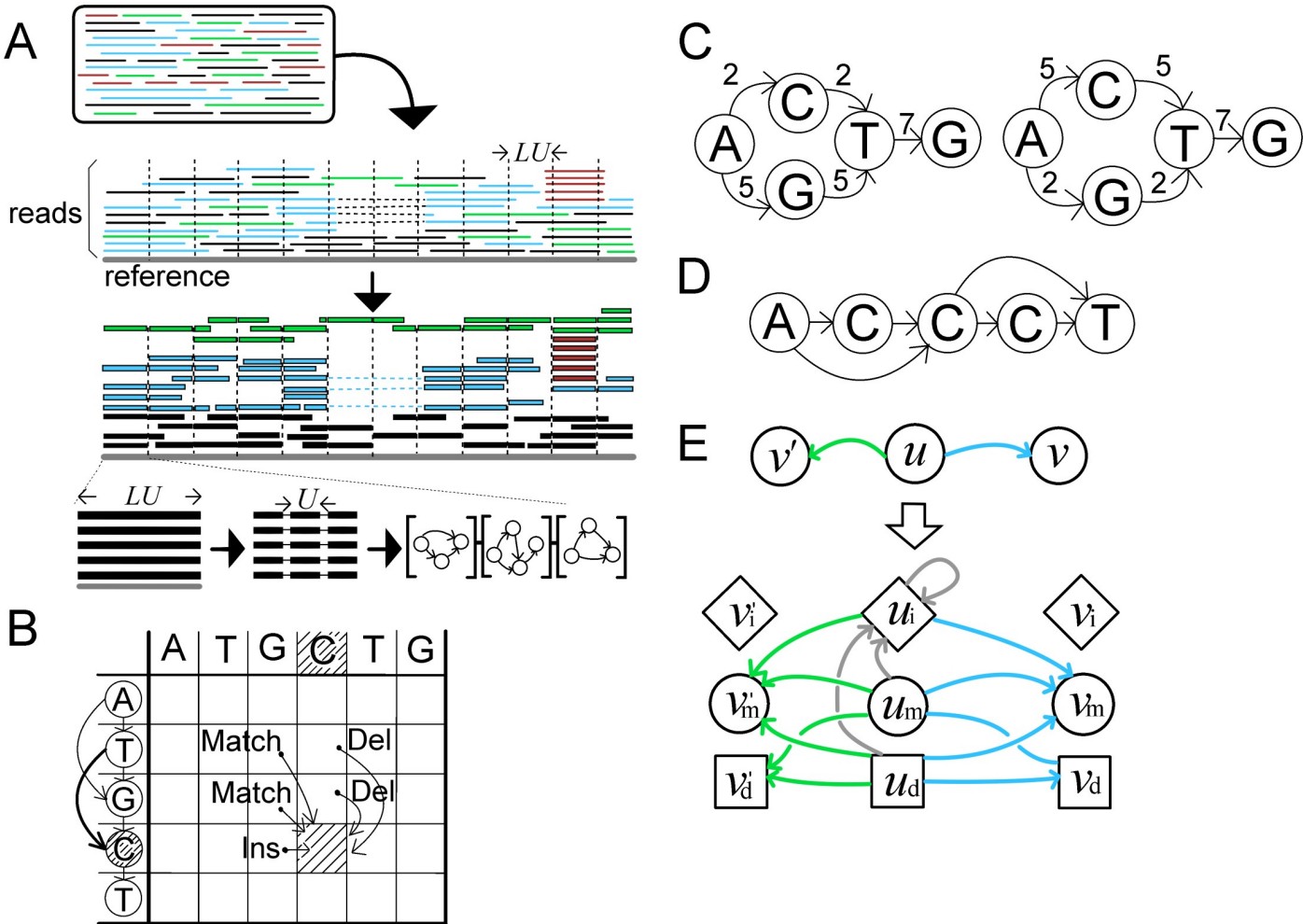

**Fig 1. Schematic illustrations of the proposed method.** (A): Overview of our workflow. *LU* is the length of a section, which we split into smaller chunks with *U*-bp length. There are reads from different multipartite structures, and we represent them by different colors (blue, green, red, and black). (B): An example of dynamic programming matrices for partial order alignments (POA). (C): Two POA graphs with the same structure but with different weights on edges. When aligning AGTG or ACTG, these two graphs display the same alignment unless we consider the weights on the edges. (D): A POA graph with a homopolymer run. Due to sequencing errors, it has two short-circuiting edges. (E): The rule to convert a POA graph into a hidden Markov model (HMM).

were converted into a sequence in a small POA graph to obtain our model. Fig 1A shows the initial steps of the workflow.

Intuitively speaking, a POA is an alignment between a nucleotide sequence and a given directed acyclic graph in which nodes represent nucleotides. The only difference between POA and classical sequence alignment is that a node in a graph may have two or more preceding nodes, making the dependency a little more complicated, as shown in Fig 1B.

Now, we define a POA between a directed acyclic graph $G = (V, E)$ where $V = \{v_1, \cdots, v_n\}$ and an $m$-length sequence $x = x_1, \cdots, x_m$ in a bottom-up manner. That is, let $G_i$ be a subgraph of $G$, in which the nodes are $V_i = \{v_1, \cdots, v_i\}$, and $x_{1:j} = x_1 \cdots, x_j$ be the $j$-length prefix of $x$. Also, let $\rho_{\text{del}}$ be a deletion penalty, $\rho_{\text{ins}}$ be an insertion penalty, and $s(v, c)$ be a match/mismatch score between the base of a node $v$ and a nucleotide $c$ for the alignment parameters. A POA score $S[i][j]$ between a subgraph $G_i$ and a prefix $x_{1:j}$ is defined based on smaller subgraphs or prefixes (Eq 1). Apparently, $S[n][m]$ is the POA score between $x$ and $G$, and it takes $O(|E|m)$

time to compute the POA score.

$$S[i][j] = \max \begin{cases} \max_{(v_p, v_i) \in E}(S[p][j-1] + s(v_i, x_j)) \\ \max_{(v_p, v_i) \in E}(S[p][j] + \rho_{\text{del}}) \\ S[i][j-1] + \rho_{\text{ins}} \end{cases} \tag{1}$$

Due to sequencing errors, POA graphs may have erroneous trailing or leading nodes. Thus, we initialized $S[i][0] = 0$ for $i \in \{0, \cdots, n\}$ and $S[0][j] = j\rho_{\text{ins}}$ for $j \in \{0, \cdots, m\}$ to allow one to leave unaligned nodes in the graph.

In the Results section, we set the insertion and deletion penalties to −3, the mismatch penalty to −4, and the match score to 3.

As with classic alignments, we can rebuild one of the alignment paths with the optimal score by back-tracing the computation of $S$. Specifically, given we follow semi-global alignment approach, we first set $i = \arg\max_i S[i][m]$, $j = m$ and update $i$ and $j$ to the values giving the maximum value of the right-side of Eq 1.

To merge a read into the graph, we introduce a new node for each mismatch or insertion operation and connect them to the adjacent nodes. For computational efficiency, we removed erroneous edges during construction of POA graph. To this end, we recorded each edge's occurrence ($e_i$) and removed erroneous edges with weight less than $\tau \times \max_i e_i$ ($\tau = 0.4$ by default) when the number of nodes reached a threshold ($1.5 \times$ maximum length of the input by default).

Overall, the pseudo-code to construct a model from reads can be found in Algorithm 2. In practice, we ran this procedure on each chunk of the reference for each cluster.

**Algorithm 2** The algorithm to construct a POA graph

```
Input: A set of sequences X = {x₁, ⋯, xₙ}
Output: A POA graph G
 1: Randomly shuffle the order of X.
 2: G ← regards x₁ as a linear graph
 3: for i = 2, ⋯, N do
 4:   Obtain an alignment between G and xᵢ by Algorithm (1).
 5:   Integrate read xᵢ into the graph G.
 6:   if | G | > 1.5 × maxᵢ | xᵢ |. then
 7:     Remove edges with weight less than 0.4 × the maximum weight of
   the edges.
 8: Return G
```

## PO-HMM to determine similarity

One of the canonical ways to define similarity between two DNA sequences is to use optimal local or global alignment scores. Likewise, one may want to use the optimal POA score to assess the similarity between a model and a read. However, this approach does not work well in practice. One reason is that, during clustering, we may have two POA graphs that differ only in the weights on their edges (as illustrated in Fig 1C), as the clustering is iteratively refined from random clustering in accordance with Line 1 in Algorithm 1.

Another reason is that, in the presence of erroneous runs of homopolymers in long reads, the POA graph will usually contain short-circuiting edges (Fig 1D). In this situation, the alignment between the graph and the query fails to segregate variable-length polymorphisms. One workaround is to assess many possible alignments using all candidate paths.

To resolve these issues and allow ambiguity in choosing nodes, we need to consider the weights on the edges and sub-optimal alignments. To this end, we converted the POA graph to

another model, the HMM. Specifically, we encoded the weights of the edges as the alignment score and defined the similarity between a model and a read by summing up the scores of all possible alignments. Although a very similar data structure was introduced in 2010 for multiple sequence alignment [26], it has not been used for clustering of long-reads. Also, while the previous model has only match and insertion states, we introduce deletion state to the HMM to handle deletion errors.

To convert a POA graph into an HMM, we substitute each node in a POA graph with three nodes representing match, deletion, and insertion states. These nodes are connected as in the original POA graph and a loop edge is added for each insertion node to allow multiple insertions. The resulting graph can be regarded as an HMM by defining the transition probability between nodes and the emission probability of each nucleotide. The partial order HMM (PO-HMM) is formally defined below (Definition 1). An example of the conversion can be found in Fig 1E.

**Definition 1 (A PO-HMM)** *Let G = (V, E) be a POA graph. Then, the state and edges of the PO-HMM G′ = (V′, E′) created from G are as follows*:

- *States: $V' = \bigcup_{v \in V}\{v_m, v_d, v_i\}$: we denote a match, deletion, and insertion as the suffixes m, d, and i, respectively.*

- *Edges: For each edge (u, v) ∈ E, we create edges $(u_\sigma, v_\tau) \in E'$, where (σ, τ) = (m,m), (m,d), (d, m), (d,d), or (i,m).*

- *Edges: For each node u ∈ V, we create edges $(u_\sigma, u_i) \in E'$, where σ = m, i, or d.*

To define the transition probability, let $W : E \to \mathbb{R}$ be weights on E. We first normalize the weight of each edge e = (u, v) by dividing the total weight of outgoing edges from the start node–i.e., $w(e) = \frac{W(e)}{\sum_{e'=(u,v')\in E} W(e')}$. The transition probabilities and emission probabilities are then defined based on these normalized weights:

**Definition 2 (Transition probability of a PO-HMM)** *Let G = (V, E) be a POA graph, w: E → [0, 1] be normalized weights on E, and G′ = (V′, E′) be the PO-HMM from G and w. Also, let $P_{\sigma,\tau}(>0)$ denote the probability of each transition $(u_\sigma, v_\tau)$ satisfying $\sum_\tau P_{\sigma,\tau} = 1$ for all σ; the sum of the probabilities of moving from $u_\sigma$ to its subsequent nodes equals to 1. Then, for each edge e = (u, v) ∈ E with normalized weight w = w(e), let*

$$w'(\sigma, \tau) = \begin{cases} 1 & \text{if } v = u \text{ and } \tau = \text{i} \\ w & \text{otherwise} \end{cases} \tag{2}$$

*Finally, we define the transition probability in G′ as*

$$Pr\{u_\sigma \to v_\tau\} = w'(\sigma, \tau)P_{\sigma,\tau} \tag{3}$$

*In other words, we use transition probability ($P_{\sigma,\tau}$) as-is for the transition between u and multiply normalized weights for the transitions between u and v. We used the "LAST-train" package [27] to estimate $P_{\sigma,\tau}$ for each dataset.*

**Definition 3 (Emission probability of a PO-HMM)** *Let G = (V, E) be a POA graph, w: E → [0, 1] be normalized weights, and $D_c^+(v)$ be the set of directed edges from v ∈ G labeled with nucleotide c. Then, the emission probabilities of the match and insertion states of the PO-HMM*

*are as follows*:

$$Pr\{c \mid v_m\} = \begin{cases} p_m & (if\ the\ base\ of\ v_m\ is\ c) \\ (1 - p_m)/3 & otherwise \end{cases} \tag{4}$$

$$Pr\{c \mid v_i\} = \sum_{e \in D_c^+(v)} w(e) \tag{5}$$

Additionally, we assume $Pr\{c \mid v_d\} = 1$, as the deletion states never result in any observation outputs.

By using the forward algorithm for a traditional HMM, we can calculate the likelihood of a given sequence. The only difference is that we have to compute the following recurrence (6) along with a topological order of the PO-HMM, as deletion transitions never change the position of the query.

Formally, let $x = x_1, \cdots, x_T$ be an observed sequence, and $G' = (V', E')$ be a PO-HMM. In addition, let $q(\sigma, t) = t - 1$ for $\sigma = i, m$ and $q(\sigma, t) = t$ for $\sigma = d$. This is the previous position of the query given the current position $t$ and state $\sigma$. Then, $F_j[v_\sigma]$, the probability to observe $x_1 \cdots x_j$ when the $j$-th state is $v_\sigma \in V'$, can be computed as Eq (6).

$$F_t[v_\sigma] = \sum_{(u, v_\sigma) \in E'} F_{q(\sigma,t)}[u] Pr\{u \rightarrow v_\sigma\} Pr\{x_t \mid v_\sigma\} \tag{6}$$

Hence, given an initial probability $F_0[v]$ for each $v \in V'$, the likelihood ($Pr\{x \mid F_0, G'\}$) is $\sum_v F_T[v]$.

## Accurate variant calling for improved clustering

We also propose another technique, variant calling, to boost the accuracy of finding better clusters. To better understand the reasoning behind this idea, suppose we have constructed models for two clusters, each corresponding to a different haplotype. As we split the reference into $L$ chunks, we have two PO-HMM arrays–$M_1 = M_{1,1}, \cdots, M_{1,L}$ and $M_2 = M_{2,1} \cdots, M_{2,L}$. The two graphs in the $l$-th chunks should differ only if they contain sites that vary between these two haplotypes. Therefore, $M_{1,l}$ and $M_{2,l}$ should be nearly the same, as the divergence rate between the two haplotypes is extremely low in our application. Nonetheless, due to sequencing errors, $M_{1,l}$ and $M_{2,l}$ may differ at some positions just by chance. Thus, the accuracy would improve if we can select which positions to use.

To this end, we introduce $L$-dimensional vector $\boldsymbol{\beta}$ such that $\beta_l \in [0, 1]$ is large when the $l$-th position likely contains variants and small otherwise. It is then straightforward to define the log-likelihood associated with each haplotype as the weighted sum of the log-likelihood values of $L$ sections.

As an example, for the $i$-th read, let $r_1, \cdots, r_L$ be the sequence of each segment of the read ($i = 1, \cdots, N$). If the read does not have any sub-sequences aligned to the $l$-th section of the reference, let $r_l$ be an empty string. Then, let the $L$-dimensional vectors $\boldsymbol{L}_1$ and $\boldsymbol{L}_2$ be:

$$\boldsymbol{L}_k = (\ln Pr\{r_1 \mid M_{k,1}\}, \cdots, \ln Pr\{r_L \mid M_{k,L}\}),$$

and the likelihood of this read on the $k$-th cluster is then defined as $\boldsymbol{\beta L}^T$, the inner product of $\boldsymbol{\beta}$ and $\boldsymbol{L}_k$. We can assess the fit between $\boldsymbol{\beta}$ and the $i$-th read by the deviation from the diagonal

line.

$$V_i = \| (\boldsymbol{\beta} L_1^{\mathrm{T}}, \boldsymbol{\beta} L_2^{\mathrm{T}}) - \frac{\boldsymbol{\beta} L_1^{\mathrm{T}} + \boldsymbol{\beta} L_2^{\mathrm{T}}}{2}(1,1) \|^2 \tag{7}$$

Thus, we can estimate an optimal value of $\boldsymbol{\beta}$ that maximizes the sum of variance–i.e., $\sum_i V_i$. Based on a straightforward calculation, we can conclude that it is equivalent to maximizing the quadratic form below:

$$\sum_{i=1}^{N} V_i = \boldsymbol{\beta} \left( \sum_i (L_1^{\mathrm{T}} L_2^{\mathrm{T}}) \left( \mathbb{I} - \frac{1}{2}(1,1)^{\mathrm{T}}(1,1) \right) \begin{pmatrix} L_1 \\ L_2 \end{pmatrix} \right) \boldsymbol{\beta}^{\mathrm{T}}, \tag{8}$$

where $\mathbb{I}$ is the $2 \times 2$ identity matrix.

Particularly, if we restrict $\|\boldsymbol{\beta}\|^2 = 1$, the optimal $\boldsymbol{\beta}$ is the eigenvector that corresponds to the maximum eigenvalue of the middle matrix of the righthand side of Eq 8.

Now, we can call the variant site between two clusters. Assume that there are $K$ clusters. Then, by comparing every pair of two clusters, we can obtain $L$-dimensional vectors such that $\beta_{k,k'}[l]$ is close to 1 when the $l$-th location contains variants between the $k$- and $k'$-th clusters and is small otherwise. We can finally derive the probability that the $n$-th read $R_n$ is in the $k$-th cluster by using these $\boldsymbol{\beta}$ PO-HMMs for each cluster $M_k$ and the fraction of each cluster $\pi_k = |\{R_i \in R \mid w_i = k\}|/N$ (Eq 13).

Overall, all steps can be found in Algorithm (3).

$$p_{n,k} = Pr\{\text{the } n\text{-th read is in the } k\text{-th cluster}\} \tag{9}$$

$$= \frac{\pi_k Pr\{R_n \mid M_k\}}{\sum_{k'} \pi_k' Pr\{R_n \mid M_k'\}} \tag{10}$$

$$= \left( \sum_{k'} \frac{\pi_{k'}}{\pi_k} \frac{Pr\{R_n \mid M_{k'}\}}{Pr\{R_n \mid M_k\}} \right)^{-1} \tag{11}$$

$$= \left( \sum_{k'} \exp\left( \ln\frac{\pi_{k'}}{\pi_k} + \ln\frac{Pr\{R_n \mid M_{k'}\}}{Pr\{R_n \mid M_k\}} \right) \right)^{-1} \tag{12}$$

$$\propto \left( \sum_{k'} \exp\left( \ln\frac{\pi_{k'}}{\pi_k} + \beta_{k,k'}(L_{k'}^{(n)} - L_k^{(n)})^{\mathrm{T}} \right) \right)^{-1} \tag{13}$$

Lastly, we note that reads supporting the same structural variations would remain in the same cluster. Therefore, if the $n$-th read supports the $k$-th structural variants, we set $w_n = k$ and never update $w_n$ in Algorithm 3 to avoid redundant recalculations.

**Algorithm 3** Clustering algorithm for a PO-HMM

```
Input: a set of reads R = {R₁, ···, R_N} and the number of clusters K
Output: the assignment of each read R_n to a cluster w_n ∈ {1, ···, K}
  1: Randomly initialize w_n for n = 1, ···, N
  2: repeat
  3:    M_k← Create a PO-HMM from {R_i ∈ R | w_i = k}.
  4:    Determine variant sites as β by calling variants from M_k and R.
  5:    p_{n,k}← calculate (13).
  6:    w_n ← arg max_k p_{n,k}
```

```
7: until the convergence
8: Return w₁, ⋯, w_N
```

## Results

In the following experiments, we used LAST version 956 [28] and LAST-split [29] for alignment and Flye [30] version 2.8 with the `--meta` option for assembly and polishing. We used 28 threads Intel Xeon Gold 5117 CPU (2.0 GHz).

### Overview of the dataset

We used one whole-genome sequencing (WGS) dataset of *A. thaliana* L*er* strains available at PacBio's official repository (https://downloads.pacbcloud.com/public/SequelData/ArabidopsisDemoData/). We also downloaded seven WGS datasets of *A. thaliana* from a previous study [31]. These strains were the Kyoto, Sha, C24, An-1, L*er*, Eri-1, and Cvi-0 strains (SRA Accession numbers:ERR3415817-ERR3415831). Additionally, as the current reference genome (master circle) is based on the mitogenomes of the Col-0 strain, we sequenced this strain anew as a control dataset (SRA Accession numbers: DRR234977). These datasets were all sequenced using the PacBio Sequel system.

We mainly used the latest assembly (GenBank ID: BK010421.1) as the reference master circle. For the L*er* and C24 strains, we also used the master circles assembled for these strains (GenBank ID: JF729100 and JF729102, respectively).

We filtered out reads from nuclear genomes from these WGS datasets, taking into account that the region from 3245K to 3511K bp in chromosome 2 of *A. thaliana* exhibits substantial similarity with some parts of the mitogenome. After filtering, we obtained 198 ± 50M bp (SD, n = 9) of long reads for each accession, presumably sequenced from the mitogenome. The average read lengths were 12.8 ± 3.4K bp (SD, n = 9), and we achieved a coverage of approximately 600 relative to the reference genome. The average error rate was approximately 13%, and the mismatch, deletion, and insertion rates were 3.3%, 3.6%, and 6.1%, respectively.

### Performance on simulated datasets

To start, we assessed our method's performance by ground-truthing with simulated datasets. First, we generated two 9K-bp templates and introduced a few (2–4) variants. Then, we made $c$ (40–70) reads from each template with varying error rates from 1% to 15% (substitution:insertion:deletion = 4:7:4) to simulate the error pattern and read length of our datasets. For each parameter, we carried out experiments 40 times with different seeds. Using Algorithm 3, we grouped these reads into two clusters. Our model decomposed the reads into corresponding structures even when there was only one variant site (Fig 2A).

To consider the case where there are more than two clusters ($K > 2$), we generated two to six templates with a 0.05% divergence rate and created reads with 15% errors from each template. The length of these templates were 9K-bp. Fig 2B and 2C show the normalized value of the left term of Eq 13 for cases with two and six clusters. Reads generated from the same template were correctly aggregated by our PO-HMM clustering.

In addition, we generated two 500K bp artificial genomes with 0.2% divergence and two structural variations that differed by 125K bp (Fig 2D). We then simulated long-read sequences using BadRead [33] with 50, 100, and 150 coverage. By clustering via a PO-HMM with $K = 3$ and assembling each resulting cluster, these structural variations were merged into the same cluster in all cases, which cannot be achieved by inputting all the reads into current assemblers (Fig 2E and 2F).

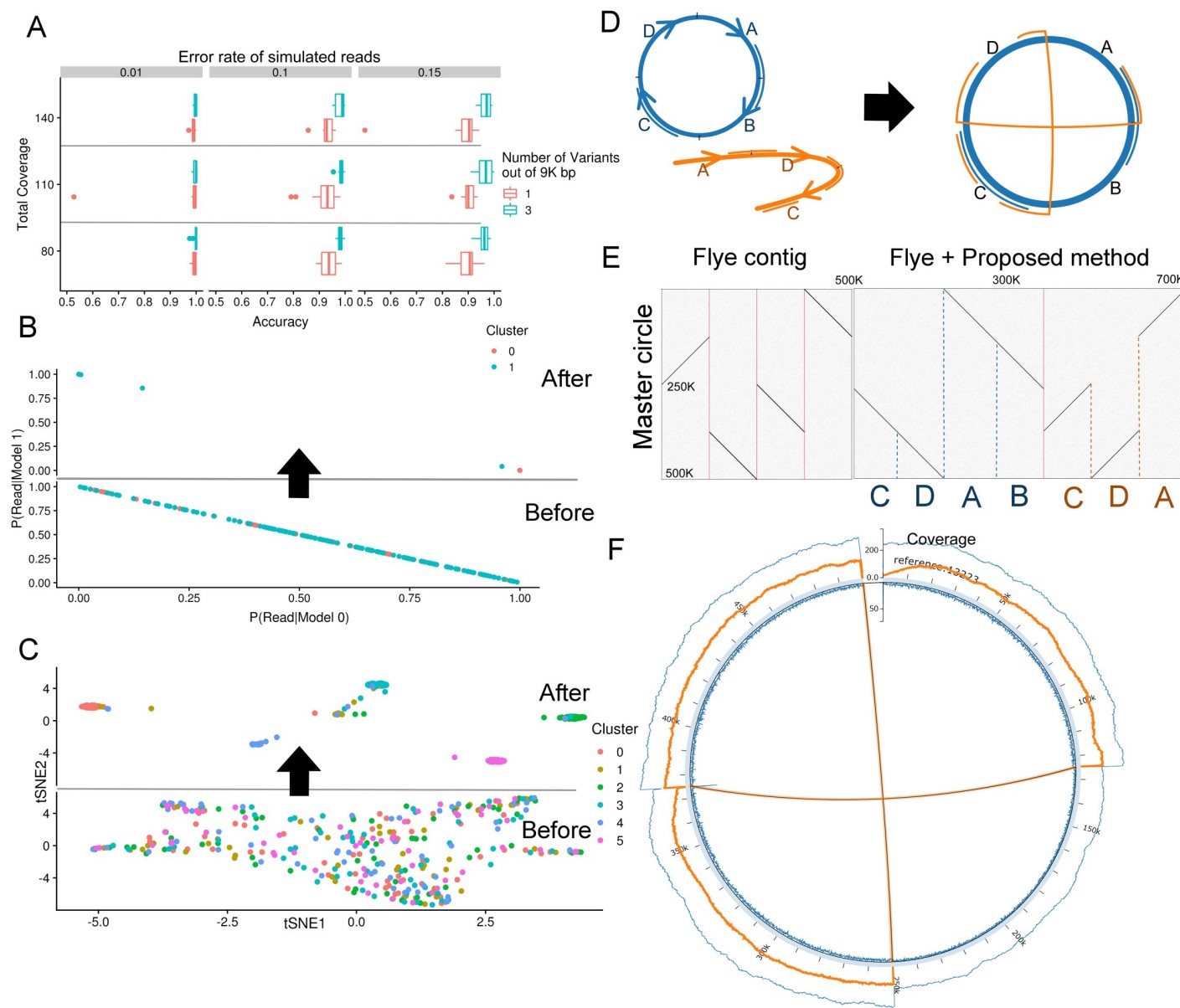

**Fig 2. Results using simulated datasets.** (A): Accuracy with respect to coverage and number of variant sites in the templates. Each box plot shows the first, second, and third quantiles of these experiments. (B): Plot of $Pr\{R_i \mid M_k\}$. We sampled 40 reads and 160 reads from the remaining template, respectively; thus, the 200 reads were unevenly distributed in the dataset. The upper figure ("AFTER") represents $Pr\{R_i \mid M_k\}$ after clustering, and the lower figure ("BEFORE") represents the values calculated based on assignments initialized randomly. (C): Plot of the posterior probabilities for six templates. We sampled 60 reads from each template. To plot the results in two-dimensional space, we used "RtSNE(https://github.com/jkrijthe/Rtsne)." 30 out of 360 reads were mis-classified. (D): Schematic illustration of two synthetic multipartite structures. The blue circle and orange curve represent a master circle and an alternative linear structure, respectively. (E): Comparison of `Flye`'s contigs produced with (right dot plot) or without (left dot plot) our proposed method. We used Gepard [32] to create these dot plots. Red vertical lines represent the boundaries of the contigs. (F): Circos plot of the mock genome. In the figure, from the outside to the inside, the blue line represents the overall coverage of the reference, the orange line represents the coverage of a minor cluster, the thick blue line represents the coordinate axis of the reference, and the number of reads that stop or start alignment at each position can be found along the inner blue line.

Lastly, we compared our method to other existing tools designed to group error-prone long reads using the dataset with 150 coverage used in the Fig 2D. Although we found no tools were designed to work on plant mitochondrial genomes, we selected three tools for clustering long reads; CARNAC-LR [22], isONclust [23], and WhatsHap version 1.0 [21]. CARNAC-LR, a

tool for *de novo* clustering for Oxford Nanopore Technologies (ONT) reads, output 39 clusters. isONclust, a tool for clustering either PacBio Iso-Seq or ONT reads, output 14 clusters as a final result. WhatsHap, a very efficient tool for haplotyping, offered `haplotag` module to clustering long-reads. It clustered reads into five clusters and there were 166 mis-classified reads out of 8923 reads. On the other hand, our PO-HMM algorithm made two clusters and there were only 112 mis-classified reads out of 8923 reads. Overall, while CARNAC-LR and isONclust output very fragmented clusters, WhatsHap and PO-HMM output higly accurate clusterings.

One downside of our method is that it requires more time than other programs. Specifically, while CARNAC-LR, isONclust, and WhatsHap took 15, 3.4, 11, minutes, respectively, to decompose the dataset, our method took CPU 110 minutes. Thus, although our approach could output accurate clustering for plant mitochondrial genomes in a resonable amount of time, it does not scale to so-called "genome scale" clustering.

## Performance on real datasets

After manual investigation of the reads indicating structural variations in the reference genome, we presumed that one minor component was present in each section of the reference in addition to the "master circle." Therefore, we set the cluster number for each section ($K$) to 3 and decomposed the reads from nine *A. thaliana* datasets. Then, we assembled each cluster to obtain putative multipartite structures for each strain.

The results for the two L*er* strains and the Kyoto strain are shown in Fig 3A–3D. In each circos plot, even though the master circles were defined as "circular," we broke the end of the contig to draw ticks.

We did not observe any apparent deletion or insertion in the master circle of the Col-0 strain, confirming that we correctly filtered out non-mitogenomic sequences from the WGS dataset. The two L*er* strains, which were treated as biological replicates in our experiment, independently exhibited a similar pattern of multipartite structures (Fig 3A and 3C), suggesting that these structures are not sequencing artifacts.

In addition to the structural variations observed in a previous study [18], we detected strain-specific multipartite structures using Algorithm 3. We observed complicated recombination patterns in the Kyoto strain (Fig 3D).

Another interesting finding is that of short (10K-bp), isolated linear structures, as indicated by black arrows in Fig 3A and 3D. We detected these linear structures in all datasets except for the Col-0 dataset, and this structure could be detected even when we used another reference master circle of the L*er* strain, indicating that this is not an artifact of the current assembly. Specifically, in the L*er* dataset from PacBio's repository, this linear structure consisted of 281 reads, and only 53 met both of the boundaries. Thus, the remaining 228 reads started from either one of the two boundaries or between boundaries. We estimated the coverage of this linear structure as 220-fold, which was 30% of the total coverage. Also, the "T-like" mark around these structures indicates that there were other structures extending in outward directions (Fig 3G).

In summary, an average of 13 clusters were obtained, ranging from 11 clusters in the Sha strain to 15 structures in the L*er* strain. Nonetheless, this may be an overestimate, as there may be clusters that do not contain any variants and are merged only with longer reads. For example, as previously reported, there were signatures of repeat-mediated recombination as shown in Fig 3A. However, we could not perfectly reconstruct these clusters, resulting in short contigs.

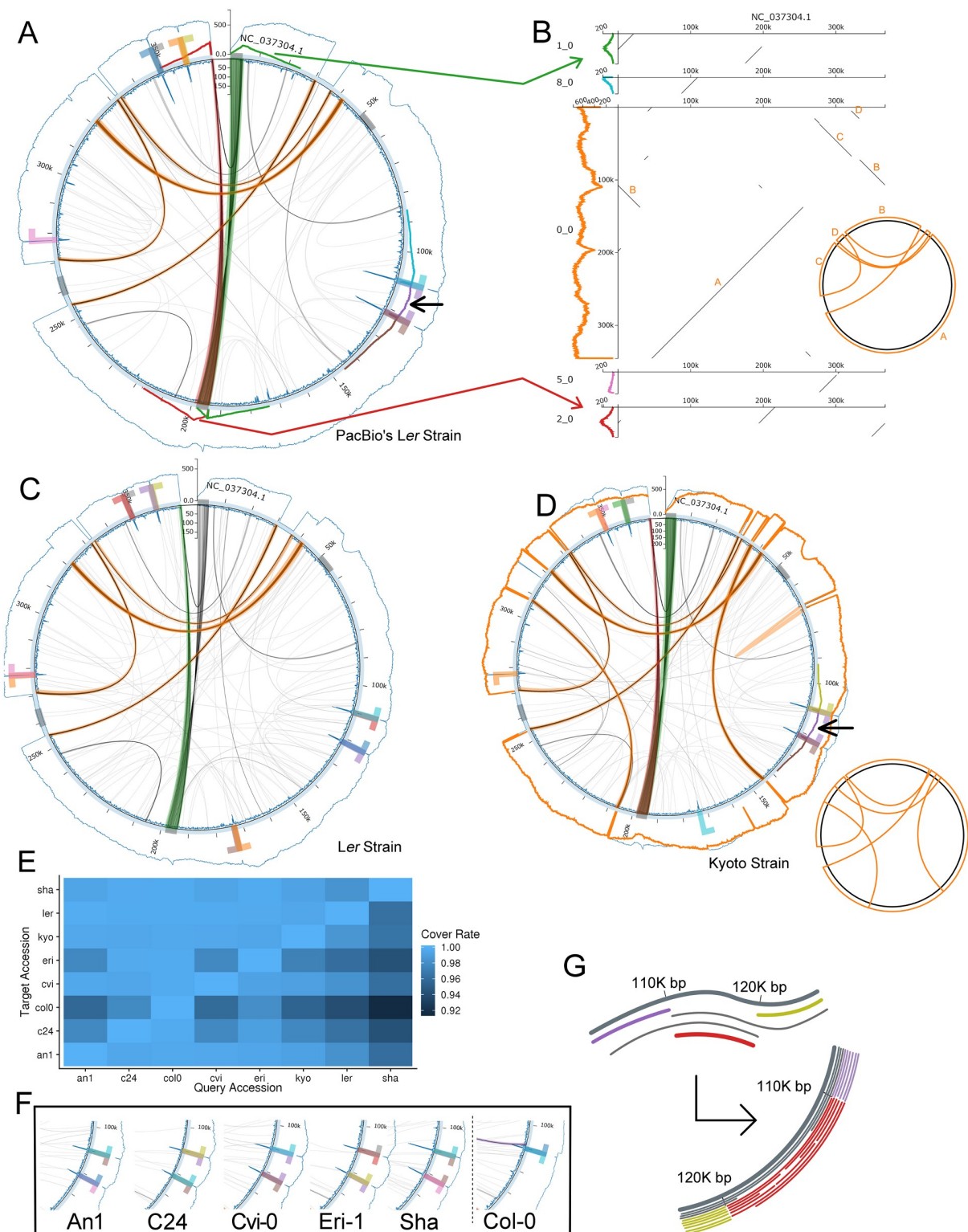

**Fig 3. Results using real datasets.** In (B) and (D), we present the schematic illustration of the largest cluster. (A): Circos plot of the L*er* dataset from PacBio's official repository. (B): Dot plots between the assembled contigs from the L*er* dataset from PacBio's official repository and the reference contig. Colored arrows indicate the correspondences between clusters and contigs. We plotted five assembled contigs from 12 clusters due to space limitations. (C): Circos plot of the L*er* dataset from [31]. (D): Circos plot of the Kyoto strain on Col-0's master circle. (E): Cover rate associated with each pair of accessions. (F): Linear structures observed in the remaining datasets–i.e., An1, C24, Cvi-0, Eri-1, and

Sha. Cropped circos plots are displayed here. (G): Schematic illustration of our hypothesis. The mitogenome is cleaved at two specific sites (upper figure), and the resulting linear structure, as represented by red lines in the lower figure, was observed.

To assess the variation among strains, we calculated the cover rate of every pair of multipartite structures. We aligned one contig onto the other and examined the length at which the reference contig was covered. We used `lastal -R00` after tuning the parameters with `last-train` to tune alignment parameters. The cover rate was 98.1 ± 1.8%(SD, n = 56), suggesting that the sequence content may be conserved despite the diverse structural variations. However, these results could be affected by the errors remaining from sequencing long reads; hence, the values may be underestimated.

## Discussion

### About our proposed method

Our proposed method, the PO-HMM, can segregate error-prone long reads from similar but slightly different genomes. Particularly, our method can be used to detect a few variant sites in error-prone long reads comprising several thousand base pairs. In other words, supporting information such as variant calling data from Illumina's short reads is not needed.

Another possible application of our algorithm is polyploid phasing of long reads [34]. However, the current implementation is not designed for genome-scale datasets and runs too slowly for use with such large datasets. We suspect that using multiple sequence alignment methods in place of the POA approach may provide much faster performance.

### About mitochondrial genomes

By applying our method, we were able to identify putative multipartite structures in nine datasets of *A. thaliana* mitochondrial genomes. The strain-specific structure found in the Kyoto strain appears to have been obtained at some point in this strain's history and then maintained within the strain. We also found a 10K-bp linear segment in all but one dataset, which we believe is a recurrent and shared structure among *A. thaliana*.

The latter finding is interesting, as there seems to be a so-called "mitochondrial-bottleneck" between each generation. As these linear structures coincided with the complementary linear structures extending in outward directions, we speculate that this structure is created *de novo* at each generation by processes such as DNA cleavage by regulatory enzymes.

As shown in the results, we failed to rebuild a few structures. This is because there were, presumably, few variations in the sequence even in the repetitive regions. Thus, some structures may differ from those in the major genome only at the structural level, which is consistent with a hypothesis stated in a previous review [19].

## Conclusion

In this paper, we introduced a new method for clustering long reads derived from very similar regions. We validated our method using simulated datasets and applied it to nine real datasets, predicting a multipartite structure for each strain. Our result not only confirmed the structural variations reported previously but also suggested co-occurrence among these structural variations.

Our suggested method can be improved further. First, The structure found, e.g. linear structure, needs to be confirmed by lab experiments. Also, it is still misleading to compare each assembled contig in a base-to-base manner, as these contigs are not completely accurate after polishing by Flye's polishing module.

Future research will focus on finding sequence features enriched in the boundaries of the detected multipartite structures. For example, as repeats are thought to mediate recombination events, we expect to find a positive relationship between the fraction of a structure and the length of nearby repeats, as implied by a previous study [35, 36].

Our PO-HMM currently does not scale to the size of genomes larger than several mega base pairs. This is because we create PO-HMMs from the reads at each loop in Algorithm 3. By solving this issue, our method would find a broader application. For example, it could be used to resolve segmental duplications in the assembly (like SDA [37]) and to distinguish haplotypes in diploid genome assembly.

## Acknowledgments

We thank for technical assistance by Yoshiko Tamura and Yu Tsuruta. Yuta Suzuki and Yoshihiko Suzuki gave us insightful comments.

## Author Contributions

**Conceptualization:** Bansho Masutani, Shin-ichi Arimura, Shinichi Morishita.

**Formal analysis:** Bansho Masutani.

**Methodology:** Bansho Masutani, Shinichi Morishita.

**Resources:** Shin-ichi Arimura.

**Software:** Bansho Masutani.

**Supervision:** Shin-ichi Arimura, Shinichi Morishita.

**Visualization:** Bansho Masutani.

**Writing – original draft:** Bansho Masutani.

**Writing – review & editing:** Shin-ichi Arimura, Shinichi Morishita.

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
