## [Decision Letter · Decision Letter 0]

16 Oct 2020

Dear Mr. Masutani,

Thank you very much for submitting your manuscript "Investigating the Landscape of Plant Mitochondrial Genomes by Long-read Sequencing" for consideration at PLOS Computational Biology. As with all papers reviewed by the journal, your manuscript was reviewed by members of the editorial board and by several independent reviewers. The reviewers appreciated the attention to an important topic. Based on the reviews, we are likely to accept this manuscript for publication, providing that you modify the manuscript according to the review recommendations.

Sincerely,

Christos A. Ouzounis

Associate Editor

PLOS Computational Biology

William Noble

Deputy Editor

PLOS Computational Biology

[LINK]

Reviewer's Responses to Questions

**Comments to the Authors:**

Reviewer #1: Dear authors:

Please refer to the following suggestions.

Title:

To my liking, talking about ‘plants’ when you only study a particular group of plants, in this case only one species, it’s rather pretentious. I know it’s something that is used by many authors, but I do not agree with this. I would change to a more appropiate title that describes the group of plants that it’s used in the study.

Abstract

1. Edit ’is not easy.’ for ‘is not feasible.’

2. Edit: ‘Specifically, to construct a sensitive read classifier, we developed a method that

exploits two classic algorithms--partial order alignment (POA) and the hidden Markov

model (HMM).’ for ‘We developed a method that exploits two classic algorithms--partial order alignment (POA) and the hidden Markov model (HMM). to construct a sensitive read classifier.’

3. Edit ‘a 15\\% sequencing-error rate and produced highly correct clustering.’ for ‘a 15% sequencing-error rate and produced correct clustering.’ It is my believed that something is correct or not, you cannot state that something is highly correct.

Author summary

1. Edit: ‘For this prepose’

2. Edit: ‘unknown structure of plant mitochondrial genome’ for ‘unknown structures of plant mitochondrial genomes’

Line 12: Edit: ‘Illumina sequencers’ for ‘Illumina sequencing technology’

Lines 34-49: The last two paragraphs of this section must be exchanged for a better organization. Minor edits must be done to accomplish this.

Materials and Methods

Line 51: Eliminate ‘To begin’

Line 72 Edit ‘longest reads’ for ‘longest read’

Line 80 Edit: ‘shows the workflow so far’ for ‘shows the initial steps of the workflow’

Line 121: ‘POA graph to a yet another model’ for ‘POA graph to another model’

Line 124 Edit ‘introduce’ for ‘introduced’

Line 127: Edit ‘let us substitute’ for ‘we substitute’

Fig. 1. Legend

1. Edit: ‘The thik line’ for ‘The thick line’

2. ‘The thik line and thin lines represent a reference and long reads, respectively’ This is confusing in the figure. Please label which is the reference and which are long reads in the figure.

3. LU and U should be defined in the legend.

Results

Fig. 2. Legend

1. Edit ‘We sampled 40 reads and 160 reads from one template and the other, respectively’ for ‘ We sampled 40 reads from one template and 160 reads from the remaining template.’

Line 228: Edit ‘Results’ for ‘Performance’

Line 239: Delete ‘the’

Lines 240-241: Delete’As can be seen’

Line 246: Edit ‘clusters’ for ‘cluster’

Line 249; Edit ‘dedicated’ for ‘designed’

Line 254: Delete ‘Similarly,’

Line 262: Edit ‘more time’ for ‘longer’

Line 267: Edit ‘Results’ for ‘Performance’

Fig. 3. Legend

1. In (D), edit ‘other’ for ‘remaining’

Line 294: Delete ’For example’

Line 298: Delete ‘Specifically,’

Line 319: Edit ‘we speculate that the Kyoto stain’ for ‘it seems that this strain’

Line 320: Edit ‘and this structure is maintained within the strain’ for ‘and this structure has been maintained’

Line 326: Edit ‘speculate’ for ‘believe’

Line 325: Edit ’by means such’ for ‘by processes such’

Conclusion

Please discuss if your method would have any application for polyploid assemblies, specially autopolyploids.

Line 338: Edit ‘However, there is room for improvement.’ for ‘There is room to improve our suggested method.’

Line 342: Delete ‘Also,’

Line 347: Delete ‘Lastly,’

Reviewer #2: The manuscript addresses an interesting problem; different from the mitochondrial genomes of animals, those of plants exist as multipartite structures that are difficult to assemble and describe using only short-read sequencing methods. The manuscript addresses this problem by proposing a new method to cluster long reads using a K-means algorithm that uses a HMM and partial order alignments to assign each long read to a different cluster. In general, the algorithms are described with sufficient detail both in the text and by using pseudocode. The underlying data and code are available from Github. The authors might need to add some extra information to the legend of Figure 1, which is somewhat vague, and need to explain better how they choose a particular value for K (the pre-defined number of clusters), as the number of existing structural variants is a priori unknown and might be very high. The manuscript might also benefit from some additional details (e.g. details on the scoring scheme used in the POA dynamic programing algorithm, or information on the starting coordinates for the back-tracing in the same algorithm, considering that the authors follow a semiglobal approach to not considering erroneous trailing nodes). Overall, I think the manuscript contains some interesting new ideas and tools to deal with the problem of cataloguing the different structural components of mitochondrial genomes in plants, a problem that is attracting much attention in the recent literature and is still waiting for solutions.

I also came across some typographical and other minor errors that the authors may want to fix:

* In the Author summary:

- it contains -> they contain/their genomes contain

- prepose -> purpose

- Appling -> Applying

- the last sentence in this section needs to be fixed too.

* Main paper:

-line 7: "a gene" is a vague expression; the authors might want to be more precise here.

- l. 8: create -> produce

-l. 51: may useful -> may be useful

-l. 75: LU? Please clarify.

- legend of figure 1: thik -> thick

- l. 102: how is the threshold selected?

l. 124: introduced

**Have all data underlying the figures and results presented in the manuscript been provided?**

Reviewer #1: Yes

Reviewer #2: Yes

PLOS authors have the option to publish the peer review history of their article (what does this mean?). If published, this will include your full peer review and any attached files.

Reviewer #1: **Yes: **Marco Cristancho

Reviewer #2: No
---

## [Decision Letter · Decision Letter 1]

1 Dec 2020

Dear Mr. Masutani,

We are pleased to inform you that your manuscript 'Investigating the Mitochondrial Genomic Landscape of Arabidopsis thaliana by Long-read Sequencing.' has been provisionally accepted for publication in PLOS Computational Biology.

Best regards,

Christos A. Ouzounis

Associate Editor

PLOS Computational Biology

William Noble

Deputy Editor

PLOS Computational Biology

Reviewer's Responses to Questions

**Comments to the Authors:**

Reviewer #1: Nice job taking care of reviewers suggestions!

Reviewer #2: All comments have been addressed. Please note that there are two different symbols for shared authorship in the cover page, suggesting that some authors are missing from the author list.

**Have all data underlying the figures and results presented in the manuscript been provided?**

Reviewer #1: Yes

Reviewer #2: Yes

PLOS authors have the option to publish the peer review history of their article (what does this mean?). If published, this will include your full peer review and any attached files.

Reviewer #1: **Yes: **Marco Cristancho

Reviewer #2: No

---

## [Editor Report · Acceptance letter]

6 Jan 2021

PCOMPBIOL-D-20-01526R1 

Investigating the Mitochondrial Genomic Landscape of Arabidopsis thaliana by Long-read Sequencing.

Dear Dr Masutani,

I am pleased to inform you that your manuscript has been formally accepted for publication in PLOS Computational Biology. Your manuscript is now with our production department and you will be notified of the publication date in due course.

With kind regards,

Livia Horvath
